# A Retrospective Study on the Role of Metformin in Colorectal Cancer Liver Metastases

**DOI:** 10.3390/biomedicines11030731

**Published:** 2023-02-28

**Authors:** Miran Rada, Lucyna Krzywon, Stephanie Petrillo, Anthoula Lazaris, Peter Metrakos

**Affiliations:** Cancer Research Program, Research Institute McGill University Health Centre, Montreal, QC H4A 3J1, Canada

**Keywords:** CRCLM, angiogenesis, vessel co-option, metformin, tumour recurrence, extrahepatic metastases

## Abstract

Colorectal cancer liver metastases (CRCLMs) have two main histopathological growth patterns (HPGs): desmoplastic (DHGP) and replacement (RHGP). The vascularization in DHGP tumours is angiogenic, while the RHGP tumours exert vessel co-option vasculature. The presence of vessel co-option tumours is associated with poor response to anti-angiogenic agents and chemotherapy, as well as a worse prognosis. Metformin has been shown to influence the progression and vasculature of tumours in different cancers. However, its role in CRCLM is poorly understood. Herein, we conducted a retrospective cohort study to examine the role of metformin in CRCLM. A dataset of 108 patients was screened, of which 20 patients used metformin. The metformin user patients did not use metformin as an anticancer agent. We noticed a significantly lower percentage of CRCLM patients with vessel co-opting RHGP tumours in the population that used metformin compared to CRCLM patients who did not use metformin. Similar results were obtained when we compared the ratio of recurrence and extrahepatic metastases incidence. Moreover, the metformin user patients had significantly higher survival outcome compared to nonusers. Collectively, our data suggest that metformin administration is likely associated with better prognosis of CRCLM.

## 1. Introduction

Colorectal cancer (CRC) is the second most lethal cancer [1] and is linked with approximately 10% of cancer-related death among men and women worldwide [2]. The development of metastatic diseases is the main cause of death in CRC patients, which over 50% of the patients will develop liver metastases (LM) during the course of their disease [3]. Surgical resection is the only chance to cure patients with colorectal cancer liver metastasis (CRCLM) [4]. However, only 15–20% of CRCLM patients are eligible for hepatic resection [5]. The unresectable patients are referred to chemotherapy combined with targeted therapies, including anti-angiogenic agents (e.g., Bevacizumab) [6,7]. However, the effect of the treatments is limited due to the acquired resistance [8].

CRCLM tumours exert two major histopathological growth patterns (HGPs) including desmoplastic HGP (DHGP) and replacement HGP (RHGP) [8,9]. The cancer cells in DHGP lesions are separated from the liver parenchyma via the desmoplastic ring [8]. However, the desmoplastic rim is absent around RHGP tumours, and the cancer cells are in direct contact with the liver parenchyma [8,10]. Moreover, the DHGP tumours use angiogenic vascularization, whereas the RHGP tumours obtain blood supply through vessel co-option vascularization [8,10]. In vessel co-opting CRCLM, the cancer cells infiltrate through liver parenchyma and hijack the pre-existing sinusoidal vessels [8,11]. Importantly, vessel co-option is associated with acquired resistance against chemotherapy and antiangiogenic drugs in CRCLM [8] and other types of cancers, such as hepatocellular carcinoma and glioblastoma [12].

Metformin is a 1,1-dimethylbiguanide hydrochloride that is extracted from the legume *Galega officinalis* [13]. Metformin is widely used to treat type 2 diabetes [14]. Metformin decreases hepatic gluconeogenesis and induces skeletal muscle glucose uptake via triggering the activation of AMP-activated protein kinase (AMPK), a master energy sensor that modulates energy homeostasis at both cellular and whole-body levels [15]. Metformin uses two different pathways for AMPK activation: (1) inhibition of NADH-ubiquinone oxidoreductase (complex I), the largest component of oxidative phosphorylation/electron transport chain, followed by reduction in the ATP/AMP ratio; and/or (2) activation of liver kinase B1 (LKB1), a protein that mediates AMPK phosphorylation and activation [15,16].

Cancer cells acquire metabolism reprogramming to obtain sufficient energy to maintain viability and build new biomass [17]. Importantly, inhibition of complex I [18], as well as induction of LKB1/AMPK pathway [19], antagonize metabolism reprogramming. Since metformin mediates both pathways [20], it has gained increasing interest as a potential anticancer agent [21,22].

Xu et al. [23] have performed a study to validate metformin repurposing as an anticancer agent and assess its role in reducing the mortality of different cancer patients. Intriguingly, the mortality was significantly lower among those cancer patients that used metformin compared to cancer patients that were not on metformin [23]. Previous studies have reported the inhibitory effect of metformin on cell proliferation, motility, and epithelial to mesenchymal transition (EMT) in cancer cells [24,25,26]. Furthermore, metformin also blocks significant signaling pathways of tumorigenesis, such as TGFβ [27] and PI3K/AKT [28] pathways. It is worth mentioning that TGFβ1 signaling pathway is significantly upregulated in vessel co-option CRCLM tumour and contributes to the expression of the proteins that mediate the development of vessel co-option, such as runt-related transcription factor-1 (RUNX1) [11].

Previous studies have proposed that low-dose metformin (250 mg/day) has potential clinical efficacy for CRC chemoprevention [29,30]. Metformin usage is also associated with better overall survival of CRC patients [31]. Metformin has multiple effects on CRC cells, including blocking cell proliferation, EMT, and motility, as well as inducing apoptosis [32]. Using CRC xenograft nude mice, Sang et al. [33] have proposed metformin as a potential antimetastatic agent in CRC, which inhibited metastases to the intestine, omentum, and renal capsule. The susceptibility of CRC cells to metformin has been correlated to different factors including the expression of miR-18b-5p, miR-145-3p miR-376b-5p, miR-718, and miR-676-3p [34]. Accordingly, the upregulation of miR-18b-5p, miR-145-3p, miR-376b-5p, and miR-718 facilitates the function of metformin in cell cycle arrest, while overexpression of miR-676-3p improves both proapoptotic and cell cycle arrest activity of metformin in CRC cells [34].

Metformin has been shown to inhibit tumour angiogenesis through various mechanisms [35]. Using metastatic breast cancer models, Wang et al. [36] have demonstrated significant inhibition of tumour angiogenesis upon treatment with metformin. Additionally, Moschetta et al. [37] have reported downregulation of hypoxia-inducible factor-1 (HIF-1) and vascular endothelial growth factor (VEGF), the key angiogenic markers in breast cancer after treatment with metformin. However, the role of metformin in the development of vessel co-option vascularization has not been studied yet.

In this manuscript, we conducted a retrospective cohort study to identify the anticancer effect of metformin in CRCLM. We assessed the distribution of the patients with different HGPs, recurrence, extrahepatic metastases, and 5-year overall survival (OS) rate upon metformin usage.

## 2. Materials and Methods

### 2.1. Clinical Data and Patient Samples

The study was conducted in accordance with the guidelines approved by McGill University Health Centre Institutional Review Board (IRB).

The data of this study were collected from all CRCLM patients who had consented to contribute to the McGill University Liver Disease Biobank research program. The presented data were collected from 108 patients who had surgical resection of their liver metastases between January 2009 and December 2020 at McGill University Health Center (MUHC), and the HGP of their tumours was determined by histopathologists. HGPs were determined after surgical resection. All patients were intended to be followed until death. Patient data was updated and reviewed through July 2022. We excluded patients with a lack of follow-up information and unknown HGPs.

The HGPs of the patients were evaluated according to international consensus guidelines for scoring the HGPs of liver metastasis [38]. Tumours with more than 50% of a specific growth pattern, i.e., DHGP or RHGP, were assigned predominately that HGP. If a patient had multiple liver tumours with different dominant growth patterns, the patient would then be designated as a patient with mixed HGP tumours.

### 2.2. Study Population

This retrospective cohort study consists of patients with pathologically confirmed CRCLM diagnoses of patients without (n = 88) and with diabetes mellitus (n = 20) (Table 1). Diabetes was defined as individuals with a self-reported history of diabetes or use of antidiabetic medications. Half of diabetic population (n = 10) was administered only metformin as antidiabetic drug, while the other half (n = 10) used metformin with at least one other antidiabetic drug before and after surgery. The metformin user patients did not use metformin as anticancer agent, while they used metformin as antidiabetic agent. All patients had surgical resection of their liver metastases between January 2009 and December 2020 at McGill University Health Center (MUHC) and their follow-up information was collected by McGill University Liver Disease Biobank research program.

### 2.3. Data Collection

Trained personnel collected demographic and clinical variables of the consented patients via medical record review using an established abstraction form. We collected information on gender, age, weight, and height, as well as administration and names of the oral antidiabetic medications. For diabetic patients, the duration of antidiabetic usage was not available.

### 2.4. Statistical Analysis

Statistical analysis was performed using GraphPad Prism software version 9.0 (GraphPad Software, La Jolla, CA, USA) software. In all instances, *p*-values of < 0.05 were considered statistically significant.

(a)The association between the two categorical groups.

We divided CRCLM patients into two major groups according to their usage of metformin, regardless of the dose and duration of metformin use and other combinational therapies they had received. The two categories were as follows: patients who did not receive (-metformin) and those who received metformin (+metformin). All metformin user CRCLM patients were diabetic. Categorical data were compared using chi-squared test. Cox proportional hazards regression model was used to estimate the hazard ratios and 95% confidence intervals.

(b)Overall Survival.

Overall survival estimates were calculated from the date of diagnosis of liver metastases to the date of death or to the date of the last follow-up. Patient data was updated and reviewed through July 2022. Kaplan–Meier analysis with log-rank tests was used to estimate survival curves and statistical significance.

## 3. Results

### 3.1. The Distribution of CRCLM Patients with Different HGPs, Recurrence, and Extrahepatic Metastases upon Metformin Usage

We established a local cohort of CRCLM from 108 patients (Appendix A) and categorized patients based on their usage of metformin; 20 (18.5%) of the patients used metformin and 88 patients (81.5%) did not use metformin. Next, we categorized both populations according to the HGPs of their tumours. Of note, the HGPs of the tumours were scored by histopathologists using 50% cut-off predominant HGPs scoring following international consensus guidelines for scoring the HGPs of liver metastasis [38]. We divided our patients into three groups of HGPs as follows: predominant vessel co-opting RHGP (44.4%), predominant angiogenic DHGP (34.3%), and mixed (21.3%), as shown in Table 1. Patients with mixed tumours are those patients who had had multiple liver tumours with different dominant growth patterns. Interestingly, we noticed a lower percentage of CRCLM patients with vessel co-option RHGP or mixed tumours in the population that used metformin compared to the population that did not take metformin (Figure 1a). Moreover, the percentage of patients with angiogenic DHGP tumours was 31% in the non-metformin population, while this ratio was 50% for the population that used metformin.

Tumour recurrence is a thorny problem in clinical tumour therapy [39]. However, limited anticancer agents have shown a significant postoperative inhibition for tumour recurrence [40]. Our data suggested that recurrence significantly increases mortality in CRCLM patients (Appendix A). To further assess the impact of metformin in CRCLM, we analyzed the recurrence incidence in CRCLM patients upon the usage of metformin. Interestingly, the usage of metformin was significantly associated with the reduction of recurrence incidence. As shown in Figure 1b, the percentage of recurrence incidence was 24% in the population of the patients that were administered metformin, whereas this ratio was significantly higher (47%) in the group that was not administered metformin.

Another factor associated with poor prognosis in CRCLM is the development of extrahepatic metastases, and it has been reported that 38% of CRCLM patients develop extrahepatic metastases [35]. Our data further confirmed these results and suggested significant reduction in the survival rate of CRCLM patients upon the presence of extrahepatic disease (Appendix A). Therefore, we decided to analyze our cohort to identify the influence of metformin on the presence of extrahepatic tumours in CRCLM. According to our data, 61% of the patients who did not use metformin have developed extrahepatic tumours (Figure 1b). However, this ratio was significantly decreased to 40% in the group of patients who were administered metformin.

While the majority of the previous studies mainly focused on metformin as an antidiabetic drug with anticancer activity, other types of antidiabetic drugs have been shown to suppress tumour progression [41]. Gliclazide [42], Sitagliptin [43], and Canagliflozin [44] are among the antidiabetic drugs that showed anticancer function. Our data showed that 50% of the CRCLM patients who used metformin were using other types of antidiabetics (Appendix A). Therefore, we decided to compare the impact of metformin and metformin combined with other antidiabetics on HGPs, recurrence, and extrahepatic incidence. Interestingly, only 20% of the patients had vessel co-option tumours in the population that used only metformin as an antidiabetic drug, while this ratio was higher (40%) in the group that used metformin combined with other antidiabetics (Appendix A). The percentage of patients with recurrence incidence in both populations was similar, at 20% (Appendix A). The ratio of patients with extrahepatic metastases was significantly lower in the population that combined metformin and other antidiabetics compared to the group that used metformin alone (Appendix A). Importantly, the difference between both groups in five-year overall survival (OS) was statistically nonsignificant (Appendix A). Taken together, the distribution of the patients suggests a significantly lower percentage of CRCLM patients with vessel co-opting RHGP tumours, recurrence, and extrahepatic metastases upon metformin usage. However, further investigations are required to verify these results and identify the molecular mechanisms underlying the role of metformin in CRCLM.

### 3.2. The Survival Rate of CRCLM Patients upon Metformin Usage

CRCLM is associated with a poor survival rate. Previous studies suggested that the median survival of CRCLM patients who underwent hepatic resection was 37.7–42.0 months after initial hepatectomy [45,46]. To examine whether metformin administration has any survival benefits for CRCLM patients, we analyzed the five-year OS in our cohort. Firstly, we compared the survival of the patients with RHGP tumours (n = 6) who used metformin to non-metformin patients with RHGP tumours (n = 42). We found a slightly better survival rate for the patients who were administered metformin compared to those who were not (Figure 2a). However, this difference was statistically nonsignificant. Similar results were obtained for patients with DHGP (Figure 2b) or mixed (Figure 2c) tumours. Next, we compared the five-year OS of CRCLM patients who were not administered metformin to those who were administered metformin. Our data demonstrated that the survival of metformin user CRCLM patients was significantly (*p* = 0.0048) higher than the rest of the patients (Figure 2d). The hazard ratio was also estimated for those using metformin individually or combined with other antidiabetics (hazard ratio (HR) = 0.8906, 95% confidence interval (CI): 0.2067–2.6274; hazard ratio (HR) = 0.2314, 95% confidence interval (CI): 0.0112–1.855, respectively). Altogether, our results proposed that metformin user CRCLM patients had significantly lower mortality than CRCLM patients who did not use metformin. 

## 4. Discussion

It has been reported that CRCLM patients with vessel co-opting RHGP tumours have the worst prognosis [8,47]. The lack of angiogenic vascularization in vessel co-option tumours is linked to their resistance to antiangiogenic agents, such as Bevacizumab [8]. Importantly, vessel co-option CRCLM tumours also showed limited response to chemotherapy [8]. Hence, impairing the development of vessel co-option tumours or converting their vasculature to angiogenic significantly increases their response to antiangiogenic agents and chemotherapy, consequently improving the prognosis of CRCLM patients.

Previous preclinical and clinical studies have proposed metformin as a potential anticancer agent [29,30]. Additionally, the usage of metformin has been associated with lower mortality in different types of cancer including colorectal, pancreas, hepatocellular, breast, and lung [48].

Tumour vascularization is crucial for tumour growth [49]. Therefore, it has become an attractive target for therapy. Tumour vascularization is divided into angiogenic and nonangiogenic [49,50]. The antiangiogenic function of metformin has been reported in various cancers, such as breast, lung, melanoma, hepatocellular, and colorectal cancer [51]. However, the effect of metformin on nonangiogenic tumour vascularization, including vessel co-option, is unknown. In the current manuscript, we found a lower percentage of CRCLM patients with nonangiogenic vessel co-option tumours upon metformin usage. Our previous publications suggested that higher levels of cancer cell proliferation, EMT, and motility [11], as well as the upregulation of TGFβ1 and PI3K/AKT pathways are essential for the development of vessel co-option CRCLM lesion [52,53,54]. Interestingly, metformin has been shown to block these pathways in various tumours [27,28]. Consequently, we postulate that the function of metformin against vessel co-option vascularization is likely mediated via attenuating the aforementioned pathways. However, this hypothesis warrants further investigation.

Tumour recurrence remains one of the major problems after hepatic resection in CRCLM and it is the main cause of death in CRCLM patients [55]. Approximately 75% of CRCLM patients experience tumour recurrence after hepatic resection [56]. Importantly, we observed a significantly lower percentage of CRCLM patients with tumour recurrence in the CRCLM population that was administered metformin compared to the CRCLM population that did not use metformin. In agreement with our findings, the usage of metformin has been linked with lower levels of tumour recurrence in patients with gastric cancer [57] and hepatocellular carcinoma [58].

Krzywon et al. [59] previously suggested that CRCLM patients with nonangiogenic tumours are more likely to develop extrahepatic metastasis. Herein, we found a smaller percentage of CRCLM patients with extrahepatic metastases upon metformin usage. Of note, CRCLM concomitant with extrahepatic metastasis is difficult to manage and associated with poor prognosis [60]. Indeed, the inverse correlation between the usage of metformin and the development of extrahepatic metastases is potentially associated with the survival benefit of metformin in CRCLM.

It has been reported that metformin reduces the serum levels of low-density lipoprotein cholesterol (LDL-C) by suppressing the transcription of proprotein convertase subtilisin/kexin type 9 (PCSK9) [61]. Importantly, we previously showed upregulation of PCSK9 in the liver parenchyma of vessel co-opting CRCLM specimens [62]. Moreover, our preclinical data suggested that using hypocholesterolemic drugs, including anti-PCSK9 (Evolocumab), significantly attenuates the development of vessel co-opting CRCLM tumours [62]. Therefore, we postulate that metformin may also affect the generation of vessel co-option tumours in CRCLM via regulation of PCSK9 and LDL-C levels. Indeed, future studies will be needed to examine this hypothesis.

Our results support prior findings that suggested the role of metformin in lowering the risk of overall mortality in cancer patients [14,51]. We hypothesize that the function of metformin in CRCLM patients is likely driven by reducing the development of vessel co-option, tumour recurrence, and extrahepatic metastases.

## 5. Limitations of the Study

The presented study has several limitations due to its retrospective design. First, the small number of patients—specifically, the patients who used metformin—may have limited proper evaluation of the correlation between HGP of CRCLM tumours and metformin usage. Second, although the data determined the usage and type of antidiabetic drugs used by CRCLM patients, they did not clarify the duration and dosage of their administration. Third, this study lacks other important information, such as the status of other comorbidities and glycemic control. Therefore, further studies are warranted to fully understand the role of metformin in CRCLM and, specifically, its impact on the development of vessel co-option tumours.

## 6. Conclusions

Our data suggested that metformin user CRCLM patients obtained better clinical outcomes. However, further studies are needed to verify our results, as well as to identify the function of metformin in CRCLM tumour resistance to antiangiogenic agents and chemotherapy.

## Figures and Tables

**Figure 1 biomedicines-11-00731-f001:**
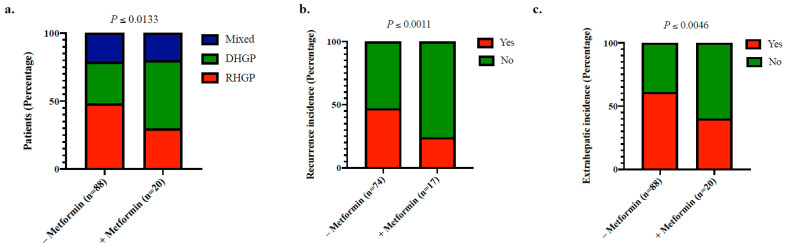
The effect of metformin on vessel co-option, recurrence, and extrahepatic metastases. (**a**) Represents the percentage of patients with vessel co-option (RHGP), angiogenic (DHGP), or mixed (RHGP and DHGP) lesions according to the administration of metformin. (**b**) Represents the association between metformin administration by CRCLM patients and the development of recurrent tumours after hepatectomy. (**c**) Shows the correlation between the usage of metformin by CRCLM patients and the development of extrahepatic metastases after surgical resection. Chi-square test was used to compare the categorical variables.

**Figure 2 biomedicines-11-00731-f002:**
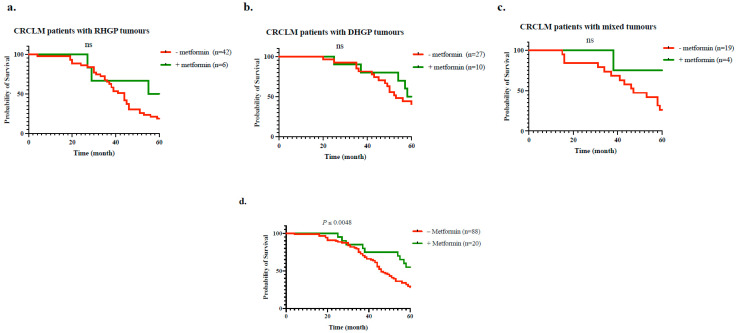
Metformin improves the prognosis of CRCLM patients. The survival analysis shows five-year overall survival rate of CRCLM patients with (**a**) RHGP, (**b**) DHGP, or (**c**) mixed tumours in the presence or absence of metformin usage. (**d**) The difference in five-year overall survival between cohorts treated with and without metformin among all CRCLM patients. Kaplan–Meier analysis with log-rank tests was used to analyze the survival curves and statistical significance.

**Table 1 biomedicines-11-00731-t001:** Demographic baseline of CRCLM patients.

	Variable	Value		
		− Metformin	+ Metformin	
			− Other Antidiabetics	+ Other Antidiabetics
**1**	**Sex**			
	Male	59	6	7
	Female	29	4	3
**2**	**Age**			
	<50 years	4	0	0
	50–70 years	45	4	5
	>70 years	39	6	5
**3**	**Body mass index category (BMI)**			
	Underweight (BMI ≤ 18.5)	0	0	0
	Normal (BMI = 18.5–24.9)	30	2	3
	Overweight (BMI = 25.0–29.9)	36	4	5
	Obese (BMI ≥ 30.0)	21	3	2
	Insufficient data	1	1	0
**4**	**Histopathological growth pattern (HGP)** **(50% predominant HGP cut-off)**			
	Replacement HGP (RHGP)	42	2	4
	Desmoplastic HGP (DHGP)	27	6	4
	Mixed	19	2	2
**5**	**Neoadjuvant before liver resection**			
	Yes	74	5	7
	No	12	5	3
	Insufficient data	2	0	0

## Data Availability

All data produced in the present study are available upon reasonable request to the authors.

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
