# Peer review of "A Retrospective Study on the Role of Metformin in Colorectal Cancer Liver Metastases"

_biomedicines, 2023, doi:10.3390/biomedicines11030731_

Round 1

Reviewer 1 Report

The topic is interesting and of novelty. However, some elements can be improved before publication:

1. Introduction should be shortened and more focused on the subject: the relation between metformin and metastasis in colorectal cancer.

2.  Material and methods section should be enlarged, so that the reader can fully understand how the cases were selected, what were the exclusion criteria. The methodology for diagnosis of hepatic metastasis and follow up  should be presented. Are the metformin group patients and non-metformin group patients paired somehow? There are many factors including age, gender, stage of colorectal cancer, associated comorbidities which may interfere with survival.

3. Results: statistical analysis should be improved.

4. Discussions: Limitations of the study should be presented.

Author Response

Reviewer #1

  1. Introduction should be shortened and more focused on the subject: the relation between metformin and metastasis in colorectal cancer.

Thanks for pointing this out. We revised the introduction and added a section that focuses on the relationship between metformin and colorectal cancer.

  1. Material and methods section should be enlarged, so that the reader can fully understand how the cases were selected, what were the exclusion criteria. The methodology for diagnosis of hepatic metastasis and follow up should be presented. Are the metformin group patients and non-metformin group patients paired somehow? There are many factors including age, gender, stage of colorectal cancer, associated comorbidities which may interfere with survival.

We agree with the reviewer, and carefully revised the materials and methods section. Also, some of these comments are answered in the revised Table 1.

  1. Results: statistical analysis should be improved.

We thank the reviewer for pointing this out. We further analyzed the data in the revised manuscript.

  1. Discussions: Limitations of the study should be presented.

We agree with the reviewer. We revised the discussion section and added limitations of the study.

Reviewer 2 Report

The short article by Rada and colleagues on the potential protective effect of metformin in CRCM patients is in a very timely and interesting topic. However, even for a Communication, there are some points that should be addressed.

The authors should be much more cautious when referring the existence of correlations (ex. Figure 1 does not show correlation, and there is no fundament to state that "metformin attenuates the development of vessel co-option") or stating that "metformin improves prognosis", when what they show is that patients with metformin intake show better prognosis. These are associations but there is no proof of causality. I would also recommend using the words "potential" or "putative" in some sentences.

Regarding methodology, the information collected from patients should be referred (demographic characteristics, etc), as well as the explanation of the HGPs scoring method. Could the authors please confirm that categorical variables analysis was performed in version 7 of GraphPad Prism?

There is a need for more information regarding Metformin usage, as the following questions arise: for how long have those patients been in Metformin before surgery? What was the dose? For how long were they still taking the biguanide, if no withdrawal? Was metformin intake related to GI disorders and/or altered motility? Are these diabetic and/or obese patients, glycemic controlled or not? Are there differences in gender distribution, based on Metformin use? These questions should be answered or highlighted as limitations and/or discussed.

The discussion should be improved. The third paragraph might be more suitable for introduction, as nothing is truly being discussed. Results can be better discussed.

Minor points:

The use of a template with numbered lines would be very appreciated to facilitate the reviewer comments.

Check list of references, as numbering shows twice from ref 17 on.

Author Response

Reviewer #2

  1. The authors should be much more cautious when referring the existence of correlations (ex. Figure 1 does not show correlation, and there is no fundament to state that "metformin attenuates the development of vessel co-option") or stating that "metformin improves prognosis", when what they show is that patients with metformin intake show better prognosis. These are associations but there is no proof of causality. I would also recommend using the words "potential" or "putative" in some sentences.

We agree with the reviewer. We carefully revised the manuscript, with focusing on using the word “potential” in describing our data.

  1. Regarding methodology, the information collected from patients should be referred (demographic characteristics, etc), as well as the explanation of the HGPs scoring method. Could the authors please confirm that categorical variables analysis was performed in version 7 of GraphPad Prism?

We thank the reviewer for pointing this out. We revised the materials and methods section and added the required information.

Regarding the data analysis question:  yes, we used categorical variables analysis.

  1. There is a need for more information regarding Metformin usage, as the following questions arise:
  2. for how long have those patients been in Metformin before surgery? What was the dose?

The patients used metformin before and after surgery, and we pointed this out in the revised manuscript (line 150). However, we do not have information about the duration and dosage of the used metformin, and we highlighted this as a limitation of the study in the study limitations section.

  1. For how long were they still taking the biguanide, if no withdrawal?

Unfortunately, we do not have information about the duration of metformin usage, while the patients continued using metformin and other antidiabetics before and after surgery.

  1. Was metformin intake related to GI disorders and/or altered motility?

No, the metformin usage was related to diabetes.

  1. Are these diabetic and/or obese patients, glycemic controlled or not?

Unfortunately, we lack information on whether they were glycemic controlled or not. We also highlighted this in the study limitations section (line 280-288).

  1. Are there differences in gender distribution, based on Metformin use?

Gender distribution in both cohorts, metformin users and non-users, is approximately similar. We showed the gender distribution of the cohorts in Table 1.

These questions should be answered or highlighted as limitations and/or discussed.

We agree with the reviewer. We listed these shortcomings in the study limitations section.

  1. The discussion should be improved. The third paragraph might be more suitable for introduction, as nothing is truly being discussed. Results can be better discussed.

We thank the reviewer for pointing this out. We revised the manuscript accordingly.

Minor points:

  1. The use of a template with numbered lines would be very appreciated to facilitate the reviewer comments.

We thank the reviewer for pointing this out. We numbered lines in the revised version.

  1. Check list of references, as numbering shows twice from ref 17.

We thank the reviewer for the keen insight, and we are deeply sorry for this oversight. We revised the references.

Reviewer 3 Report

Title: A retrospective study on the role of metformin in colorectal cancer
liver metastases

Authors: Miran Rada, Lucyna Krzywon, Stephanie Petrillo, Anthoula Lazaris, Peter Metrakos

Summary:

The subject of this "communication" is very interesting. However, there are major gaps in the study described, both in its description and in the available data. The missing information needs to be filled in, and if it is not available, the limitations of the present results need to be discussed in detail.

Several major points are listed below:

1) In my opinion, the study presented has significant shortcomings. For example, I lack answers to the following questions:

(a) How should the average data be comparable when only 20 of 108 patients were taking metformin? This information should be clearly pointed out in the results and discussion, since this is a very unequal group.

b) The division of the groups described in the results is confusing. Was a comparison made between patients taking/without metformin for each of the three groups?

c) In Material and Methods, the design of the study should be explained in detail.

d) Information is missing on whether the patients studied were taking metformin anyway as an antidiabetic or whether they received it specifically for this study as an anticancer agent. This would be relevant in the abstract, in the results and also in Table 1.

e) Information on duration and dosage of metformin is lacking, so one cannot derive a recommended course of action from the data.

f) The limits of data interpretation need to be clearly stated.

2) The reference list needs to be revised:

(a) Starting with reference 17, the references are numbered twice. From line 43, the numbering is also shifted.

(b) In line 198 the references have to be linked correctly.

(c) In the line there is a discrepancy between the named author and the bibliography.

d) Please add the following additional references for introduction/discussion:

doi: 10.1038/s41598-022-06587-9,

doi: 10.3390/cancers13020188,

doi: 10.3390/curroncol28020134,

doi: 10.3390/biom9010016.

Author Response

Reviewer #3

1) In my opinion, the study presented has significant shortcomings. For example, I lack answers to the following questions:

(a) How should the average data be comparable when only 20 of 108 patients were taking metformin? This information should be clearly pointed out in the results and discussion, since this is a very unequal group.

We agree with the reviewer that our study has several shortcomings including the size of the cohorts and we pointed this out in the study limitations section.

  1. b) The division of the groups described in the results is confusing. Was a comparison made between patients taking/without metformin for each of the three groups?

We are deeply sorry for the confusion. We revised the manuscript and clarified the study design more clearly (line 177-180).

  1. c) In Material and Methods, the design of the study should be explained in detail.

We thank the reviewer for pointing this out. We revised the materials and methods section and added more details.

  1. d) Information is missing on whether the patients studied were taking metformin anyway as an antidiabetic or whether they received it specifically for this study as an anticancer agent. This would be relevant in the abstract, in the results and also in Table 1.

We thank the reviewer for pointing this out. We revised the manuscript and clarified the reason for metformin usage by CRCLM patients.

  1. e) Information on duration and dosage of metformin is lacking, so one cannot derive a recommended course of action from the data.

We agree with the reviewer, and we pointed this out in the study limitations section.

  1. f) The limits of data interpretation need to be clearly stated.

We agree with the reviewer, and we added the study limitations section.

2) The reference list needs to be revised:

(a) Starting with reference 17, the references are numbered twice. From line 43, the numbering is also shifted.

(b) In line 198 the references have to be linked correctly.

(c) In the line there is a discrepancy between the named author and the bibliography.

We thank the reviewer for the keen insight, and we are deeply sorry for this oversight. We revised the references of the manuscript and fixed the mistakes in the reference list.

  1. d) Please add the following additional references for introduction/discussion:

doi: 10.1038/s41598-022-06587-9,

doi: 10.3390/curroncol28020134,

doi: 10.3390/biom9010016.

doi: 10.3390/cancers13020188,

We thank the reviewer for the suggested references, and we added them to the revised manuscript. 

Round 2

Reviewer 1 Report

the article have been revised. I have no further queries.

Author Response

Thank you so much!

Reviewer 2 Report

The authors have improved their manuscript and answered to some of my concerns.

However, there are still important points to be improved before publication. I would recommend having the support of a statistics specialist.

Figure 1 does not show the effect of metformin, but rather the distribution (frequencies) based on metformin intake. As mentioned in the previous round of review, the authors should be more careful with causality, as there is no proof of the true effect. In b., there is no association; in c., there is no correlation.

The specific statistical test performed should be disclosed in every figure legend.

The authors have now included novel sub-analysis of their original data depending on the use of other oral antidiabetic; this is only making the analysis to loose power, because the size of the sample, which was already small, decrease even more. Nevertheless, it is all in supplementary, so it is not clear why making that point. Moreover, I was not able to find figures S1a-d.

If the authors want to share supplementary data in a very raw format (excel file), I would recommend to be at least more rigorous in the nomenclature, avoiding to use both active ingredient (drug name) and brand (ex.: januvia instead of sitagliptin).

I understand the authors aimed for a Communication and not a Full Article, however, a proper Discussion is still missing.

Adding the limitations of the study is appreciated.

Author Response

1. Figure 1 does not show the effect of metformin, but rather the distribution (frequencies) based on metformin intake. As mentioned in the previous round of review, the authors should be more careful with causality, as there is no proof of the true effect. In b., there is no association; in c., there is no correlation.

We agree with the reviewer. We rewrote the sentences that mentioned the effect, association, or correlation in the revised manuscript. In the revised manuscript, we mainly showed and discussed the results as a distribution between two groups of patients including metformin users and non-users.

2. The specific statistical test performed should be disclosed in every figure legend.

We added the statistical test methods in the figure legends.

3. The authors have now included novel sub-analysis of their original data depending on the use of other oral antidiabetic; this is only making the analysis to loose power, because the size of the sample, which was already small, decrease even more. Nevertheless, it is all in supplementary, so it is not clear why making that point. Moreover, I was not able to find figures S1a-d.

We are sorry that you could not find figure S1, we believe the issue was from the journal because we have uploaded figure S1.

We believe showing these data is important, as it further analyzes the distribution of HGPs, recurrence, extrahepatic metastases, and survival of the patients upon metformin usage, whether the metformin used individually or with other antidiabetics. Additionally, we think that these results will be important for the investigators who will be studying the role of antidiabetics as anticancer agent in future, specifically for treating vessel co-option tumours.

4. If the authors want to share supplementary data in a very raw format (excel file), I would recommend to be at least more rigorous in the nomenclature, avoiding to use both active ingredient (drug name) and brand (ex.: januvia instead of sitagliptin).

We agree with the reviewer, and we changed the names.

5. I understand the authors aimed for a Communication and not a Full Article, however, a proper Discussion is still missing.

We added another section to the discussion.

6. Adding the limitations of the study is appreciated.

We thank the reviewer for acknowledging our effort.

Reviewer 3 Report

The authors have addressed all points of potential criticism and each suggestion made by the reviewer adequately and in detail.

Author Response

Thank you so much!